# Hazard Identification of Hydrogen-Based Alternative Fuels Onboard Ships

Erin van Rheenen *, Evelien Scheffers , Jesper Zwaginga  and Klaas Visser

Department of Maritime and Transport Technology, Delft University of Technology, 2628 CD Delft, The Netherlands; e.l.scheffers@tudelft.nl (E.S.); jesper.zwaginga@tudelft.nl (J.Z.); k.visser@tudelft.nl (K.V.)
* Correspondence: e.s.vanrheenen@tudelft.nl

**Abstract:** It is essential to use alternative fuels if we are to reach the emission reduction targets set by the IMO. Hydrogen carriers are classified as zero-emission, while having a higher energy density (including packing factor) than pure hydrogen. They are often considered as safe alternative fuels. The exact definition of what safety entails is often lacking, both for hydrogen carriers as well as for ship safety. The aim of this study is to review the safety of hydrogen carriers from two perspectives, investigating potential connections between the chemical and maritime approaches to safety. This enables a reasoned consideration between safety aspects and other design drivers in ship design and operation. The hydrogen carriers AB, NaBH$_4$, KBH$_4$ and two LOHCs (NEC and DBT) are taken into consideration, together with a couple reference fuels (ammonia, methanol and MDO). After the evaluation of chemical properties related to safety and the scope of the current IMO safety framework, it can be concluded that safety remains a vague and non-explicit concept from both perspectives. Therefore, further research is required to prove the safe application of hydrogen carriers onboard ships.

**Keywords:** safety; hydrogen carriers; hazard identification; marine transportation; marine engineering; alternative fuels; methanol; ammonia

## 1. Introduction

The shipping sector, transporting 80% of the volume of international trade in goods, was responsible for approximately 2% of the total GHG emissions in 2021 [1]. The IMO has set a target to reduce GHG emissions by 50% of the 2008 emissions in 2050 to stimulate the industry in becoming more sustainable [1]. However, reality shows that fossil fuels such as HFO and MDO, which still emit high concentrations of pollutants, are still widely used [2]. Alternative fuels are essential in order to reach the set targets [1]. The potential of alternative fuels is also recognized by the shipping community. Therefore, the maritime industry and academics perform extensive research on alternative fuels such as LNG, methanol, hydrogen and ammonia [2,3]. These fuels are considered cleaner than current options, depending on the production and conversion process. Of the more prevalent alternative fuels, only hydrogen can be classified as emission-free, but it unfortunately has a very poor energy density, which makes it too low to be relevant for shipping purposes [3,4].

Instead of pure hydrogen, hydrogen carriers might be used to overcome this disadvantage. These are defined as circular, hydrogen-rich, liquid- or solid-phase materials from which hydrogen can be liberated on demand. These hydrogen carriers are often considered to be safe alternative fuels [3,5–10]. Such claims imply that they can be safely integrated in the power and propulsion systems onboard ships. However, the exact definition of what safety entails is often lacking, both for hydrogen carriers as well as for ship safety. Occasionally, the literature takes hydrogen carrier safety into account, but the application of safety on ships is missing or not specified with sufficient detail [11,12]. On the other side, the MSC of the IMO is involved in defining overarching safety requirements for ships and

their fuels [13]. However, these safety prescriptions are currently limited to conventional fuels and an explicit design philosophy for safety is lacking. Therefore, this study aims to review the safety of hydrogen and investigate potential connections between the chemical and maritime approaches to safety.

The considered hydrogen carriers and reference fuels are introduced in Section 2, followed by an identification of their hazards in Section 3. The second viewpoint, the maritime perspective and approach of safety, is summarized in Section 4. The last two sections integrate the first three sections by matching the chemical properties of hydrogen to maritime safety aspects and by providing a conclusion in Sections 5 and 6, respectively.

## 2. Hydrogen Carriers

Hydrogen carriers store hydrogen atoms through chemical bonding or physical adsorption inside other substances [14]. An extensive review of different hydrogen carriers for shipping purposes has been performed before by [12]. The review looked at energy density, storage, safety and handling, the dehydrogenation process, the TRL level and the recycling process and found a set of hydrogen carriers that can be regarded as having potential to be used on ships. Table 1 gives the energy densities of the considered hydrogen carriers, including reference fuels: MDO, ammonia and methanol. The latter two are commonly considered as alternative fuels onboard ships [2,3]. Section 2.1 discusses the first three carriers of Table 1 (AB, $NaBH_4$ and $KBH_4$) in further detail. Section 2.2 explains the LOHCs NEC and DBT and Section 2.3 deals with the reference fuels ammonia and methanol. As MDO and HFO have very similar GHS labels, they will be referred to in this paper as MDO [2,15,16].

Furthermore, because this paper approaches these considered fuels from a safety perspective, the technological details of the considered fuels are left out. Instead, the reader is referred to [9] for an overview of methanol and ammonia and to [12] for an overview of the hydrogen carriers. These hydrogen carriers are all categorized as (relatively) safe [7,11,17].

**Table 1.** Theoretical volumetric and gravimetric energy densities of hydrogen carriers for explicit application onboard ships.

| Carrier | MJ/kg | Source (MJ/L) | MJ/L | Source |
|---|---|---|---|---|
| AB | 23.52 | [8] | 14.4 | [18] |
| $NaBH_4$ (hydrolysis) | 25.56 | [19] | 27.34 | [19] |
| $KBH_4$ (hydrolysis) | 17.76 | [20] | 20.78 | [20] |
| LOHC: NEC | 6.98 | [21] | 6.63 | [21] |
| LOHC: DBT | 7.44 | [11] | 7.0 | [22] |
| Ammonia | 21.12 | [23] | 11.5 | [23] |
| Methanol | 15.12 | [10] | 11.88 | [10] |
| MDO | 29 | [24] | 30 | [24] |

### 2.1. Borohydrides and AB

The borohydrides $NaBH_4$ and $KBH_4$ are characterized by similar chemical properties, as they are solid powders, which slowly oxidize in air [7]. Furthermore, boranes (such as $NH_3BH_3$) is also considered in this section as it is similar in reaction type, storage and handling, even though it is not officially a borohydride. These substances react exothermically with water to form hydrogen, through a process called hydrolysis [7,17,20], which is shown for $NaBH_4$ below.

$$NaBH_4 + (2 + x)\,H_2O \rightarrow 4H_2 + NaBO_2 \cdot xH_2O \tag{1}$$

The borohydrides have medium TRLs and the first ship using $NaBH_4$ is currently under construction [25]. $NaBH_4$ and $KBH_4$ are extremely similar substances regarding reaction type, storage and handling, but since sodium is lighter than potassium, the energy density of $NaBH_4$ is higher [20]. The spent fuel presents a major problem for both borohydrides, as

it is much heavier than the initial fuel. This is because the -BH$_4$ changes into a -BO$_2$, and oxygen is heavier than hydrogen. Furthermore, depending on the temperature, it might become a hydrate. The amount of water stored in the remaining product is fuel-specific, with NaBH$_4$ storing more water than KBH$_4$. Additionally, water removal through drying is easier for KBH$_4$ [26]. Nevertheless, the end weight of the end product at the same temperature is still very similar.

AB (NH$_3$BH$_3$) is a combination of ammonia and boron, which has similar characteristics as borohydrides [7]. Its main advantage is an extremely high energy density [7]. If NH$_3$BH$_3$ is hydrolysed, ammonia and boric acid are formed [7,27], as it reacts with water:

$$H_3NBH_3 + 3H_2O \rightarrow NH_3 + B(OH)_3 + 3H_2 \qquad (2)$$

Because of this, the substance is more difficult to use than the other borohydrides, as the spent fuel also contains gaseous ammonia [27]. This needs to be either broken down further or stored separately, but can also be used as a fuel in a combustion engine when using a pilot fuel. Overall, NH$_3$BH$_3$) is a very promising fuel, if its complete energy potential can be used [12]. However, its exact TRL is unknown, which indicates a lower readiness.

### 2.2. LOHCs

Liquid organic hydrogen carriers (LOHCs) are usually in liquid form. The release of hydrogen from LOHCs is an endothermic process, requiring energy and elevated temperature. There are three LOHCs that are generally considered, namely DBT, NEC and toluene [9,11,28]. However, toluene is not considered here because of safety concerns regarding a low flashpoint, high carcinogenicity and the difficult and energy-costly dehydrogenation process [9]. DBT, on the other hand, is easy to store and easy to handle [11]. It has a very high TRL and is currently in preparation as being an energy carrier in the shipping industry [29]. The only main drawback is the high temperature required for the hydrogen release, of 543–583 K [11,30]. NEC is also considered by [11] to be a very promising carrier, due to its lower dehydrogenation temperature (of 453 to 543 K). It is also considered to be non-toxic and very easy to handle. Its main drawbacks are the TRL level of only 3, its solid form at ambient conditions and low TRL level, of only 3 [11]. LOHCs in general do not have very high energy densities, but are considered for shipping purposes because of their easy use as they have very similar properties as crude oil [11,12]. They do not need much additional infrastructure due to their liquid, oil-like form and are deemed to be safe [11]. It is also assumed that this helps in public acceptance [11].

### 2.3. Ammonia and Methanol

Methanol is considered to be CO$_2$-neutral and ammonia CO$_2$-free, depending on the production method [3]. As methanol can be made from CO$_2$, it can be viewed as CO$_2$ neutral because the same amount of CO$_2$ can be captured and released during formation and conversion back to electric or mechanical energy [9,10]. Methanol is promising for the shipping industry because it has a high energy density and is easy to store, as it is liquid under ambient conditions [9]. Ammonia too can be made from hydrogen, and this production process is fully developed because ammonia is also used as fertilizer [3,23,31]. Ammonia has a very high energy density, but it has to be stored under either temperature slightly below freezing (263 K) or at slightly elevated pressure (1000 kPa) [3,9].

## 3. Hazard Identification of Hydrogen Carriers and Selected Reference Fuels

The discussed hydrogen carriers are considered to have potential for maritime usage based on their energy densities and TRL. This section only describes the potentially hazardous consequences as retrieved in the GHS and the more detailed safety data sheets. It does not aim to provide a hazard analysis, as this asks for a probabilistic representation of the likeliness of each hazard, which is strongly dependent on the specific context. The properties of substances in certain environments, such as flammability, toxicity and reaction to water, are established through so-called hazard assessments [32]. These provide the

background for the GHS labels and the safety data sheets. Both the hydrogenated and hydrogenated forms of hydrogen carriers have to be taken into account. Additionally, some of the hydrogen carriers react with water, splitting up into different substances. The hazards of these substances are also taken into account here, as it can be expected that, in a spill, alternative fuels come into contact with water. The hazards of thermolyzed substances are considered as well because of the temperature of hydrogen fires (2400 K) [33]. This chapter provides a comprised hazard identification of the hydrogen carriers mentioned in the previous section, starting with two LOHCs and followed by the borohydrides, ammonia and methanol. For each carrier, the relevant classes out of six applicable GHS classification categories are covered in the following order; flammable, acute toxic, health hazard, corrosive, irritant and environmental hazard. Of these, flammable represents a physical hazard that can cause physical damage, environmental hazards are hazardous to the aquatic environment, and the other four present dangers to human health [34]. Section 3.7 summarizes the results, followed by Section 3.8 with two hazards that are applicable to all hydrogen carriers.

### 3.1. Ghs Classification and Qualitative Research

It should be noted that the following work is completely qualitative. This is mainly because the GHS only has a limited way of defining the hazards associated with each category. There are a total of nine pictograms, which all can be classified as either warning or danger [34]. Additionally, two pictograms, those of 'irritation' and 'environment' are always only classified as 'warning' [34]. This thus results in only 16 ways of categorization of the dangers associated with substances. Within these major categories, there are subcategories, but these do not quantify the hazard, only specify it, e.g., 'flammable solid' [35]. The GHS also does not give exact values, such as toxicity potential indicator or lethal doses [34]. Similarly, for corrosivity, the GHS identifies two hazards under corrosion hazards—H290 (corrosive to metals) and H314 (causes severe skin burns and eye damage)—but refrains from examining other forms of material corrosion since they are not classified by the GHS. This means that possible hazard propagation due to material corrosion is also not covered. All materials that are mentioned here as corrosive are accompanied by code H314, thus being corrosive to the skin and eyes. Consequently, we acknowledge that substances may still possess the ability to corrode other materials or exhibit mild corrosivity that falls below the prescribed threshold value. However, this is generally not considered hazardous by the GHS, thereby falling outside the scope of this paper. It should be noted, however, that GHS statements are not static and are prone to modifications with the advent of new information. For example, hazard classifications have not been performed for all hydrogen carriers. For some of the hydrogen carriers or substances that are a result of reactions that happen with these carriers, there is no classification at all; it is unknown how dangerous these substances are. On the other hand, some hydrogen carriers such as NEC have been thoroughly assessed using a hazard assessment [11,36]. This adds to the uncertainty of the already relatively blunt form of qualifying hazards using the GHS symbols. So, the assessment of hazards, including the influence of alternative fuels on other materials, is paramount for the integration of alternative fuels in general, including ammonia, methanol and pure hydrogen. All in all, current available knowledge and use of GHS classification results in a very broad division of detail of the hazards of hydrogen carriers. This is why the categorization here in this paper is performed in a qualitative way. Because of the lack of quantification, it should be noted that comparison between two different hydrogen carriers is extremely difficult, even though they may have the same GHS pictograms, there can still be a lot of difference in the exact qualitative outcome of the comparison.

### 3.2. Hazard Identification of LOHC: Dibenzyltoluene

DBT is commonly thought of as a promising LOHC. For safety purposes it has as its main advantage that there is no significant byproduct produced during dehydrogenation [11]. This means that when DBT is on board, only the hydrogenated and dehydrogenated versions

of DBT itself have to be taken into account. DBT is a rather safe substance, as it has low volatility [6], low flammability [10] and is not carcinogenic [6]. Despite it being generally categorized as safe by authors [6,10,11], DBT forms a health hazard as it can be deadly if it is swallowed and reaches the airways, and because it may damage the unborn child [37]. Furthermore, as dibenzyltoluene is a polycyclic aromatic hydrogen carbon, there is evidence that it can cause breast cancer [38]. Even though it is biodegradable [39], it is a possible environmental hazard [11] and is a long-term hazard toxic to aquatic life [37]. More exactly, there have been studies that show that DBT is very toxic to aquatic life in the long term, but not acutely toxic in the short term [40]. Of the hydrogenated version of DBT, so-called perhydrodibenzyltoluene, there are little data available [11]. We could not find a safety sheet for this substance. This is mainly because DBT itself has been used as a heat transfer oil for technical applications already, thus being mature and requiring safety data sheets [36]. Safety measurements are thus still required. Therefore, DBT is a safe substance to use on ships, as long as it does not leak, as it is an environmental hazard to aquatic life.

### 3.3. Hazard Identification of LOHC: n-Ethylcarbazole

Contrary to DBT, when NEC is dehydrogenated, small byproducts are produced [11]. However, these are not discussed as only 2% of NEC degenerates and there are multiple economic and technological reasons to further limit the production of byproducts. NEC is less toxic than DBT or MDO [11,41]. Consequently, NEC falls into the lowest toxicology level possible [41]. Nevertheless, it is classified according to GHS as a strong irritant as well as a long-term hazard toxic for aquatic life and the environment [42,43]. The hazard assessment of [41], shows that it is not easily biodegradable (dehydrogenated NEC) to poorly biodegradable (hydrogenated NEC). There is limited information on the hydrogenated version of N-ethylcarbazole, known as either perhydro-n-ethylcarbazole or 9-Ethyldodecahydro-1H-carbazole [44]. According to a vendor, it has to be stored in a closed container in a dry and cool place and it is an irritant [45], just like the dehydrogenated version [42]. If further hazard assessment confirms this, NEC is mainly hazardous due to its irritant, corrosive and pollutant properties, and leakage should be prevented.

### 3.4. Hazard Identification of Borohydrides

$NaBH_4$ is considered as flammable, corrosive, acute toxic, irritant and health hazard according to the GHS symbols [46]. It is categorized as flammable due to the release of hydrogen when in contact with water, while the substance is not very flammable itself. According to the safety data sheet, it is combustible but hard to ignite [47]. $NaBH_4$ is corrosive to skin, causing severe burns [46]. However, it is not necessarily corrosive to metals. $KBH_4$ has a similar safety data sheet, but is in general less dangerous than $NaBH_4$. It is corrosive to skin, flammable (when in contact with water) and toxic for people [48]. Just like $NaBH_4$, it causes severe skin burns. Therefore, both the borohydrides should be stored away from people and the environment.

#### 3.4.1. Spent Fuel

The spent fuel of the borohydrides consists of several products [26,49]. For $NaBH_4$, mainly $NaBO_2$ and $NaB(OH)_4$ remain, both of which are classified as irritant and health hazard, but only on the warning level [50,51]. For $KBH_4$, mainly $KBO_2$ and $KBO_2 \cdot 1/3H_2O$. $KBO_2$ has very similar properties to $NaBO_2$, as it is an irritant and health hazard on the warning level [52]. Both $KBO_2 \cdot 1/3H_2O$ and $NaB(OH)_4$ split up in water, as they are salts, with the restproducts being sodium and potassium ions and $B(OH)_4^-$ (aq) (Tetrahydroxyboranuide) [49]. Tetrahydroxyburanuide, or tetrahydroxoborate, does not have a safety sheet of its own, however it has very similar properties to borax [53], which is a health hazard and is categorized as a danger in the reproductive category [54].

### 3.4.2. Thermolysis

Next to the hydrolysis of the borohydrides, they can also be thermolyzed. This happens at elevated temperatures, such as temperatures occurring in a hydrogen fire. At these elevated temperatures, the borohydrides decompose in the following products [55]: sodium, boron, borol and sodium hydride. Elemental sodium, the major component, is extremely dangerous, as it is flammable and may ignite spontaneously when coming into contact with water [56]. Additionally, sodium is corrosive to human skin, causing severe burns and damage. Both pictograms are classed as 'danger' [56]. Boron, on the other hand, is only toxic if swallowed [57]. Other substances, such as sodium hydride (flammable, spontaneous ignition when contact with water) and borol (flammable, toxic, health hazard) are dangerous as well [58,59]. The decomposition temperature of $KBH_4$ is 700 K and that of $NaBH_4$ is 693 K [20]. Thermolysis of $KBH_4$ forms similar products as those of $NaBH_4$, such as potassium, boron and potassium hydride [20]. Potassium is flammable and corrosive to skin, and can, just like sodium, ignite when it comes into contact with water [60]. Potassium hydride is again very similar to sodium hydride, being flammable, ignites when in contact with water and corrosive to the skin [61].

### 3.5. Hazard Identification of AB

AB is a flammable solid (danger level) and an irritant for skin, eye and respiratory tract [62]. Hydrogen release from AB can be performed in two ways, thermolysis and hydrolysis [7]. Hydrolysis is very similar to the hydrolysis of sodium and potassium borohydride and visible in Equation (2). The boric acid formed during hydrolysis immediately ionises in water into $B(OH)_4^-$. Thermolysis starts at temperatures of 373 K [63]. All in all, the following substances can be created when using AB:

- $NH_4^+$ (Hydrolysis reaction, relatively stable reaction)
- $B(OH)_4^-$ (Hydrolysis reaction, see $NaBH_4$, relatively stable reaction)
- $NH_2BH_2$, aminoborane, extremely unstable, oligomerises easily (Thermolysis, 100 °C)
- HNBH, iminoborane, extremely unstable, oligomerises easily (Thermolysis, 150 °C)
- Borazine (Thermolysis, result of oligomerization)

### 3.5.1. Hydrolysis

The main products of hydrolysis are $NH_4^+$ and $B(OH)_4^-$, the latter of which has been discussed in Section 3.4.1 already. For the hydrolysis it is known that AB is relatively stable and needs a catalyst to have dehydrogenation when it comes into contact with water, thus this might be less likely to happen [63]. Ammonium ($NH_4^+$) is a nitrogenous ion that is the conjugate of ammonia [64]. Dependent on the pH level of the solution, the balance of the equation goes to ammonia (high pH) or ammonium (low pH) [65]. As the sea has a general pH of 8.08–8.33 [66], it can be assumed that both ammonium and ammonia are present after the hydrolysis reaction. Ammonia will be discussed in more detail in Section 3.6, as it can also be used as an alternative fuel by itself. Ammonium will be discussed here. Ammonium is commonly used in households, as cleaning agent but also for personal body hygiene [64]. Ammonium is corrosive to the eyes and causes skin irritation [64]. It causes an overabundant growth of aquatic plants, as it is a nutrient for them [65]. However, it can also become toxic when it transforms into ammonia. Ammonium itself is thus not considered to be toxic to aquatic life [64,65].

### 3.5.2. Thermolysis

During thermolysis, dangerous gasses, such as borazine, are released, which is why hydrolysis is the most opted version of hydrogen release from AB [7,27,63]. However, as spontaneous thermolysis starts at 373 K to 423 K, it is relevant for safety purposes [63]. AB is not necessarily a stable substance due to this low thermolysis temperature. Thermolysis produces several products. The definition and accompanying names of these products of [27] are used here, which may differ from those of [67,68]. After the first step at 373 K, aminoborane is formed, but there is very limited information on this substance. However,

it is mentioned to be highly reactive [27]. At 423 K, aminoborane decomposes into HNBH (iminoborane) and hydrogen [27]. Iminoborane (also known as Boraneimine), $BNH_2$, is formed during an endothermic reaction. Both are mentioned to be similar to ethylene acetylene [27]. The final result from the thermolysis reactions, iminoborane, is thermodynamically unstable [69]. It oligomerizes easily into borazine ($H_3B_3N_3H_3$) [27,69,70]. Borazine is in its own a very dangerous substance, being flammable and corrosive to skin [71]. It reacts very violently with water and may ignite upon contact [71]. Thus, if the thermolysis steps were to happen, producing borazine, which in its turn can then ignite upon contact with water, triggering new thermolysis reactions and creating a positive feedback loop.

### 3.6. Hazard Identification of Ammonia and Methanol

Ammonia and methanol are both widely produced and shipped all over the world [9]. Depending on the energy release mechanism, releasing energy from ammonia either produces $NO_x$ or pure nitrogen. Ammonia is a flammable gas, although it requires preheating before ignition can occur [72]. The storage is difficult because it is gaseous at ambient conditions, corrosive to metals and skin, acutely toxic if inhaled and very toxic to aquatic life [3,72,73]. The first three issues are in the 'danger' category, the environmental hazard (toxicity to aquatic life) is considered to be a warning [72]. Even though ammonia is less reactive than conventional fuels, the toxicity is seen as a major issue for its use as a future fuel [73,74]. Ammonia is usually stored under low temperature or slightly elevated pressure, so that it becomes a liquid. This way of storing it can cause dangerous, toxic clouds, which spread over a large area [73]. Another issue is that this toxic cloud can come into contact with water, resulting in the formation of a layer of toxic $NH_4OH$ [2]. Thus, the toxicity of ammonia is a major issue.

Methanol is a volatile substance and liquid under ambient conditions [2]. It is extremely flammable and can combust under ambient conditions [75]. Additionally, it can drop down to the source of its ignition and ignite again [2,75], because it is heavier than air. Furthermore, methanol burns invisible, making it hard to detect in bright daylight [75]. Furthermore, methanol is acutely toxic and a health hazard [75]. It should not be touched as it is toxic in contact with the skin. If swallowed or inhaled, it also causes damage to organs and low exposure over the long term is dangerous [75]. For marine life, on the other hand, methanol is not classified by the GHS as toxic [75].

### 3.7. Overview of Hazards of the Hydrogen Carriers and Reference Fuels Based on the GHS System

Table 2 gives an overview of the known GHS symbols of the hydrogen carriers. Hydrogenated and dehydrogenated versions of the carriers are taken into account. "D" stands for "Danger" which implies a more severe form of that said hazard than a "W". L is the lowest category, implying only a label has to be used, with no additional warning or danger marking [35]. It should be noted that only one of them can appear, always the most dangerous one is chosen.

**Table 2.** GHS categories for hydrogen carriers, ammonia, methanol and MDO. * indicates that this hazard only occurs after contact with water and ^ when the substance has been thermolyzed.

| Hazard Class | DBT | NEC | NaBH$_4$ | KBH$_4$ | NH$_3$BH$_3$ | NH$_3$ | Methanol | MDO |
|---|---|---|---|---|---|---|---|---|
| **Flammable** | | | D * | D * | D | | D | |
| *Acute Toxic* | | | D | D | D * | D | D | D |
| *Health Hazard* | D | | D | D * | D * | | D | D |
| *Corrosive to skin* | | | D | D | W */D ^ | D | | |
| *Irritant* | | W | W | W * | W | | | W |
| **Environmental Hazard** | W | | | | W * | W | | L |

It should be noted here that even though MDO is categorized as flammable in the fourth category, this is not classified as a hazard in the GHS system [15]. Table 2 gives the

danger and warning levels as advised by [35]. It can be seen from the table that the LOHCs have the least hazards and the borohydrides (and AB) have the most hazards.

The results of Table 2 can also be shown graphically. This way it becomes immediately clear which substances have similar properties. Figure 1 shows these. In this figure, all dangers and warnings are taken into account, including those that only occur when the substance is in contact with water or heated. This means that Figure 1 shows the worst case scenario of all substances. It should be noted here again that this is a qualitative figure, as this paper only looks at potential consequences. Figure 1 shows that MDO has a large amount of safety hazards. It can also be seen that methanol follows the same pattern as MDO for acute toxicity and health hazard. AB touches all possible limits in the chart, as environmental hazards and irritants only have the warning level, making it clearly the most dangerous substance. The borohydrides follow the exact same pattern, because the reactions with water are also taken into account. Comparing this to Figure 1a, it can be seen that there only the borohydrides are corrosive to skin.

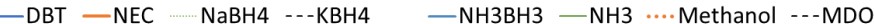

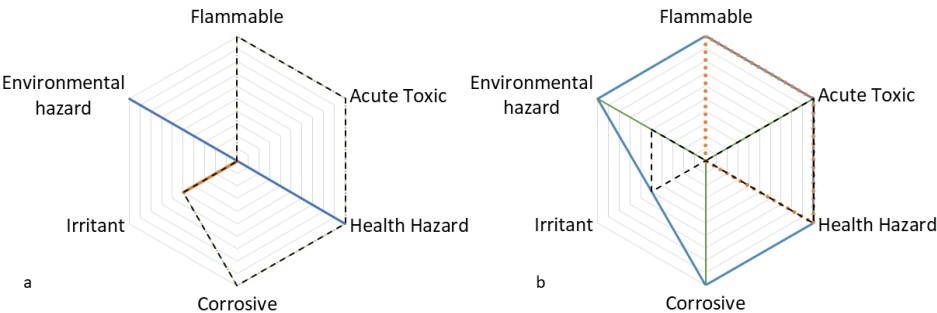

**Figure 1.** (**a**) Spiderweb of safety hazards of LOHCs (DBT and NEC) and borohydrides (NaBH$_4$ and KBH$_4$); (**b**) Spiderweb of safety hazards of AB (NH$_3$BH$_3$), ammonia, methanol and MDO.

### 3.8. General Hydrogen Carrier Hazards

Besides the specific hazards discussed above, several general hazards exist due to the presence of hydrogen, of which two will be covered. First, as all considered fuels are able to release hydrogen (including methanol and ammonia through cracking), hydrogen fires present a substantial danger. Second, substances that are released in the air, for example in the case of a hydrogen fire, can extend the affected area far beyond the boundaries of a ship.

#### 3.8.1. Hazard of Hydrogen Fire and Explosion

When hydrogen is released, it can easily ignite, because of its broad flammability range of 4–75% [76]. Historically, the main cause of incidents involving hydrogen are mechanical failures, which can result in several reactions: jet fires, flash fires, explosions and atmospheric dispersion [77]. Ignition usually results in a jet flame, which is a long flame from the point of ignition that can even occur after an initial explosion [78,79]. A flash fire is a fire without an explosion, which is started by an external ignition source [78,80]. Such fires are likely to happen when liquid hydrogen is spilled [33] and its main property is a flame that travels back to the leak [80].

A considerable hazard of hydrogen is its explosive properties. These have been studied well, especially under practical circumstances, using liquid or pressurized hydrogen [33,76–78,80]. Ref. [33] found a realistic and conservative relation between the size of the fireball and the mass of hydrogen involved:

$$D_{hmsc,c} = 19.5 \times m_{H2}^{1/3},\tag{3}$$

which shows that already small amounts of hydrogen can cause very big fireballs. Due to the very low density of hydrogen, these fireballs are almost never completely circular and the hydrogen can travel rather far before ignition (up to 30 m from the source) [33,81]. The detonation limit of hydrogen is 18.3–59% and explosions mostly occur due to over-pressure in concealed spaces. Possible reason for over-pressures are the expansion of cryogenic hydrogen after vaporizing (850 times increase) and poorly handled compressed hydrogen [82]. Venting is often used as a solution, but according to [83], the explosion peak over-pressure only decreases with larger vent sizes. Regarding the size of explosion, a study by [84] on the Fukushima accident investigated the increase in damage 30 m from the reactor core for larger amounts of hydrogen. They showed the pressure wave from 10 kg of hydrogen results in minor structural damage, 80 kg results in major structural damage and 200 kg could result in partial demolition. An example of damage closer to the detonation is given by [76], who studied a hydrogen explosion in an ammonia plant. In this case, 10–20 kg of hydrogen was discharged and only about 3.5–7 kg of exploded. Nevertheless, this still resulted in concrete blocks weighing 1.2 metric tons being moved up to 16 m [76]. Despite the power of the explosion, the heat flux is limited. For example, [33] shows that for a tank rupture test (35.7 MPa, 1.64 kg of hydrogen), the first-degree burn exposure (with the marker 'pain'), is only 5 m, with an exposure time of 2.8 s. Despite the fact that hydrogen has an adiabatic flame temperature of 2403 K, this exposure is not enough to cause second- (or even third-) degree burns [33]. Consequently, even though hydrogen explosions have high power, their thermal doses are low, meaning the explosions are the most dangerous on a ship.

### 3.8.2. Aerosol Hazard Identification

For a substance to become dangerous outside the vessel boundaries, it would have to be either propelled away by an explosion, or be in the form of aerosols or other small particles. These are able to become "airborne", or spreading through the air by following air flows [85]. The exact size, density and humidity of the droplet are important parameters defining the spread [85]. For example, [85] showed that droplets with diameters smaller than 10 μm could be more dangerous than larger ones, as these stay in the air for longer periods of time, while droplets with diameters of 10 and 20 μm fell down after about 350 s. Besides this, small droplets can also spread rapidly, covering several meters in a matter of seconds. This principle is visualized in Figure 2, where different-sized particles where released from the same location and simulated over 35 s. This confirms that substances which form particles smaller than approximately 10 μm, have a danger of becoming airborne, which could result in travel over large distances. This airborne hazard is relevant for cryogenic fuels, which create vapour clouds after spillage, and ammonia, which creates toxic ammonia clouds when a substantial amount comes into contact with water [9]. Consequently, it is evident that not only local effects, within the boundary of the ship, should be taken into account for these fuels.

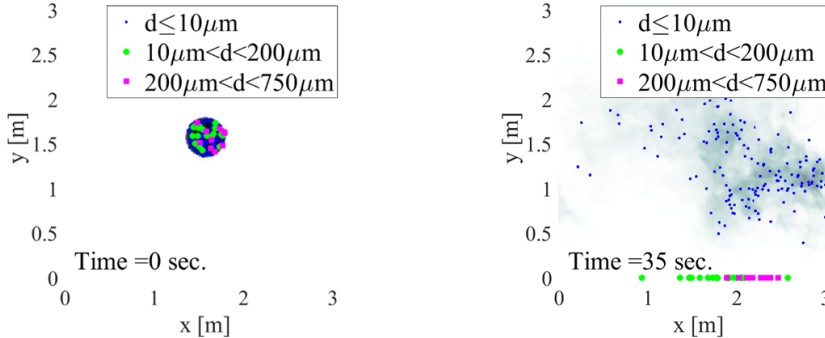

**Figure 2.** Flow of droplets over time; fluctuation level 0.05 m/s RMS for three different droplet sizes; own work with use of DNSLab [86].

## 4. Current Safety Approach in the Maritime Industry

This chapter investigates the current maritime regulations and guidelines to understand the influence of the chemical properties on the design of and the application on ships. The main philosophies and regulating structures are discussed to provide a better understanding on the origin and approach of safety regulations. The IMO (a UN specialized agency) is the main international organization with regards to the regulation of international shipping and navigation for safety, for vessel-source pollution and for maritime security purposes. In this paper, the IMO conventions and guidelines are used to determine how safety is currently defined onboard ships. We have categorized safety in the following three safety aspects: First of all, the safety of life on board, such as safety of the crew and passengers. Secondly, there is the safety of the ship itself, the structural integrity of the ship. The third and final point is the safety of the ecosystem surrounding the ship. The surroundings of the ship are here defined as the area which a fuel (spill) can influence. Section 4.1 discusses the regulations regarding these safety aspects. Another categorization can be made based on the regulation philosophy: prescriptive standards versus goal-based standards—the latter is described in Section 4.2.

### 4.1. IMO Safety Regulations

The three aforementioned safety aspects, of life on board, of the ship itself and of the ecosystem surrounding the ship, are covered in a range of conventions and regulations. Historically, these rules are based on incidents and written from a prescriptive perspective. For example, SOLAS was adopted in 1914, motivated by the Titanic disaster in 1912 [87]. The first aspect, safety of life of crew members and passengers, is an inherent goal of convention SOLAS, which covers minimum prescriptive standards for the construction, equipment and operations of ships [88]. In SOLAS, safety is defined as "protection from danger, risk or injury, in the context of non-intentional events, such as accidents or events cause by human error" [89]. Furthermore, the MLC determines "the minimum working and living standards for all seafarers on those ships" [90]. The MLC, the authority on occupational safety, has defined a hazard as the inherent potential to cause injury, harm or damage to a seafarer's health. This includes many sources, for example intrinsic (chemical) properties, situations, potential energy, the environment or human influence. Passenger safety, on the other hand, is mainly focused on communicating the relevant SOLAS requirements effectively to passengers [91]. For example, on passenger ships (ships designed to carry over 12 passengers), the regulatory safety aspects most relevant to crew are the general fire fighting system and the possible escape routes in case of a casualty [92]. Therefore, passenger safety does not seem to be directly influenced by chemical properties of potential fuels within a regulatory perspective.

The second safety aspect, ship safety and structural integrity, is part of the IGF Code. The goal of this code is "to provide criteria for the arrangement and installation of machinery for propulsion and auxiliary purposes, using natural gas as fuel, which will have an equivalent level of integrity in terms of safety, reliability and dependability as that which can be achieved with a new and comparable conventional oil fuelled main and auxiliary machinery" [93]. The current IGF code is primarily prescriptive, even though a goal-based approach is followed. It has been amended using interim guidelines for alcohol fuels [94], while ammonia and hydrogen are still in the progress of being added [95]. Therefore, it is not yet applicable to all alternative fuels. Most importantly, it is only applicable to more prevalent alternative fuels, but not to the hydrogen carriers.

The main convention on environmental safety is MARPOL [96]. It applies to two different situations: during daily operation and in case of pollution by incidental causes. The Annexes of MARPOL mainly focus on water pollution, such as pollution by oil, noxious liquid substances, harmful substances carried by sea, sewage and garbage. Annex VI, however, aims to prevent air pollution, limiting GHG, $SO_x$, $NO_x$ and PM emissions.

As mentioned in Section 3.8.2, aerosols with a small diameter can spread over significant areas depending on the wind speeds. This is only a risk for fuels that are gaseous

at ambient conditions and have lower or similar densities compared to air [2]. Therefore, ammonia is the only fuel discussed in this study for which this risk applies. IMO considers the toxicity one of the key safety aspects in using ammonia, posing dangers for both life on board and nearby personnel [97].

### 4.2. Goal-Based Standards (GBS)

MSC introduced a goal-based philosophy into the SOLAS Convention in 2002 [98]. The goal-based design approach aims to provide greater freedom in developing technical solutions and accommodating different standards. Goal-based regulation does not specify the means of achieving compliance but sets goals that allow alternative ways of achieving compliance in contrary to conventional 'prescriptive' standards [99]. A five-tier system was introduced to develop the GBS of which the first three (goals, functional requirements and verification of compliance criteria) are performed by IMO. Classification societies and industry are responsible for the fourth and fifth tier, respectively. This creates a responsibility distribution amongst all parties involved in safe and future-proof shipping.

The main goal of tier I is applicable to all new ships and as follows: "ships are to be designed and constructed for a specified design life to be safe and environmentally friendly..." [100]. Within Tier 1 of the GBS, the following objective relate directly to safety [100]:

1.  Safe and environmentally friendly means that the ship shall have adequate strength, integrity and stability to minimize the risk of loss of the ship or pollution to the marine environment due to structural failure, including collapse, resulting in flooding or loss of watertight integrity.
2.  Environmentally friendly also includes the ship being constructed of materials for environmentally acceptable dismantling and recycling.
3.  Safety also includes the ship's structure being arranged to provide for safe access, escape, inspection and proper maintenance.

### 4.3. Implications of IMO Safety Regulations on Hydrogen Carriers

The IMO has recognized two main codes that deal with safety of gas and low flash-point substance as cargo (IGC) and fuels (IGF). However, as shown for LNG by [101], codes like these often take many years to be adopted, while safety rules lack clear technical justification or limit application. Furthermore, as these codes focus on the application to specific substances, some hydrogen carriers are not covered. Therefore, the MSC, the IMO committee that creates legislation in the field of maritime safety and security [102], has approved 'Interim Guidelines for the safety of ships using fuel cell power installations'. This guideline states that safety, reliability and dependability of alternative fuels should be in line with general conventions [103].

Classification societies play a crucial role in the maritime industry by converting codes such as the IGC and the IGF into prescriptive rules, as well as overseeing compliance. Notably, some classification societies have recently published rules designed to provide guidance for alternative fuels like liquefied natural gas (LNG), ammonia, and methanol [104,105]. However, it should be emphasized that these regulations are not yet universally applicable to all hydrogen carriers.

Upon a thorough analysis of the current International Maritime Organization (IMO) regulations and conventions, it is evident that numerous safety aspects have been addressed based on lessons learned by past incidents. Serious steps have been made towards GBS, slowly moving away from prescriptive standards. However, the definition of safety for life on board, structural integrity and ecology remains somewhat elusive. The regulations are either not applicable on hydrogen carriers (prescriptive) or very broad (GBS). As such, the context in which alternative fuels such as hydrogen carriers are to be integrated into remains not strictly defined.

## 5. Influence of Hazards Accompanied by Alternative Fuels on The Approach to Safety on Ships

In this section, the connection between hazards and on board applications will be made, categorized in line with the previously introduced IMO safety aspects (ecological, public health, crew and ship). Potential mitigation strategies are not taken into account, as these can be major design choices. Public health is not included in this section, as its only applicable to one fuel, ammonia. In this section, the identified safety aspects as described in Section 3 are categorized in the context of application onboard ships. Section 5.1 goes over the application of safety hazards that specifically influence the life onboard ships. Section 5.2 looks at the influence of safety hazards on the structural integrity of ships and Section 5.3 regards the influence of the substances on the (aquatic) environment.

### 5.1. Possible Influence of Alternative Fuels on Life on Board

As mentioned in Section 4.1, there are differences between the safety aspects of life of crew and passengers. For passengers, there are additional requirements regarding communication. The safety of all life on board is related to hazards, as these can cause injury or damage to health. Current fuels like MDO are already dangerous substances. Figure 3 shows a comparison of the hazards when compared to MDO.

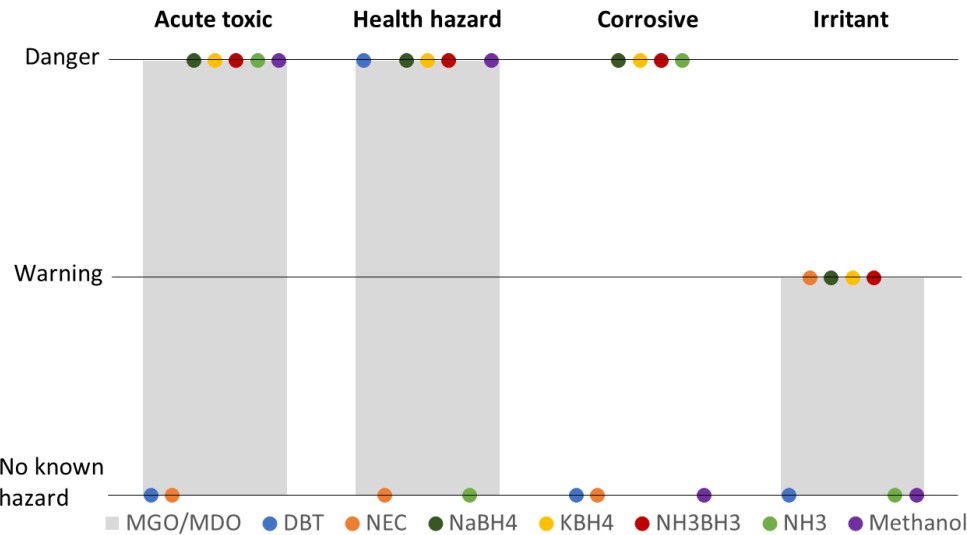

**Figure 3.** Danger and warning levels of alternative fuels relevant for the health of crew and passengers, compared to those of MDO.

All substances that are labeled corrosive are corrosive to the human skin, as can be seen in Section 3. As visualized in Figure 3, some fuels have similar hazards, such as DBT, $NaBH_4$, $KBH_4$ and the products of hydrolysis of $NH_3BH_3$: boric acid and borax. The latter is also a rest product of hydrolysis of $NaBH_4$ and $KBH_4$. These substances are all capable of possible damage to the fertility or the unborn child. However, it is not clearly defined what this GHS statement means. This is problematic, as impact on reproductivity and pregnancies can be impacted even after exposure and the impact of chemicals on women is less known [106]. For example, the ILO protects women that are pregnant or nursing by requiring measurements to ensure the workplace is safe [106]. This would most likely mean additional measurements for DBT, $NaBH_4$, $KBH_4$ and AB.

It can be seen that there are three substances on a similar safety level to the health of the crew compared to MDO. These are DBT, NEC and methanol, which will be discussed here first. Ammonia follows and after that the borohydrides will be discussed as they have similar properties. Lastly, AB is discussed.

### 5.1.1. Methanol and LOHCs DBT and NEC

Methanol has a similar acute toxicity and level of general health hazard compared to MDO. Even the exact health hazards are very similar, namely both should not be swallowed or inhaled and cause skin irritation. Additionally, both cause damage to the organs. The main difference here is that MDO is also suspected of causing cancer, so methanol is less hazardous to life.

DBT and NEC are both less hazardous to life than MDO, since NEC only is a strong irritant. The main difference between the irritant properties of NEC and MDO is that NEC also causes eye irritation. NEC causes skin irritation and may cause respiratory irritation, but does not seem to be harmful if swallowed [42,44]. As these irritation levels are only on warning level, NEC can be categorized as more safe to the crew of the ship than MDO. DBT is slightly more hazardous to the crew than NEC, as it poses a health hazard on danger level. This is because DBT may be fatal if swallowed, which is the same warning as MDO (H304). Additionally, as DBT can endanger the unborn child [37], while MDO does not have this warning. However, DBT is not considered acutely toxic or an irritant, making it in general more safe to the crew, especially if they are all male. This would mean that the above-mentioned three fuels are very likely to comply with the regulations of the MLC, as the hazards are less than those of MDO.

### 5.1.2. Borohydrides

A substance that scores a "danger" hazard on all levels is $NaBH_4$ [46]. This is because it is extremely corrosive to the skin but also acutely toxic when swallowed. Additionally, it may damage fertility or the unborn child, which thus causes a health hazard. This substance should thus not come into contact with the crew of the ship at all. Unlike the other substances mentioned previously, $NaBH_4$ is a powder and thus less likely to move around through the air on the ground. This, combined with the fact that it is only dangerous when touched or swallowed, results in that in the event of a spill, $NaBH_4$ is less likely to pose a danger to the crew than ammonia (which can spread freely, being a gas at ambient conditions). Thus, a $NaBH_4$ spill will be dangerous for the crew locally, but most likely less dangerous for passengers on board. $KBH_4$ has the same properties for acute toxicity and corrosiveness to skin, but it only becomes a health hazard and an irritant when it comes into contact with water. This is due to the formation of potassium metaborate and borax, which are a health hazard to the unborn child [52,54]. Additionally, potassium metaborate is an irritant, but only on the warning level, as it causes eye irritation [52]. Thus, $KBH_4$ is in the end less hazardous than $NaBH_4$.

### 5.1.3. AB

AB only has a warning for being an irritant, so it should not be swallowed, touched or inhaled [62]. In itself, AB can thus be considered as less hazardous than MDO, as MDO is classified as warning in the hazard class irritant and also being dangerous in acute toxic and health hazard class. However, AB does become extremely dangerous when in contact with water, as it produces ammonia [27]. However, this reaction is considered to be relatively stable and requires catalysts [63]. However, when this happens, AB forms ammonia as well as borax, which have been mentioned previously. Another danger is that AB can decompose under elevated temperatures (up to 423 K). This results in the compound borazine. Borazine is corrosive to skin on a danger level [71]. To sum this up, AB itself is less hazardous than MDO. However, when heated, it becomes corrosive to skin (which MDO is not) and when in contact with water it becomes an irritant, health hazard and acutely toxic due to the formation of ammonia and borax.

### 5.1.4. Ammonia

Ammonia is widely known that to be acutely toxic at inhalation[72–74]. Besides this, it is only classified as corrosive to the human skin [72]. Because of the few, but dangerous hazards, ammonia is hard to compare to MDO. Both have different hazards influencing

human health and both may lead to death if entering the airways [15,72]. MDO has additional hazards, such as health hazards, which ammonia does not have. This does not, however, make ammonia more safe than MDO. The hazards are expressed differently, as the risks are different. Additionally, ammonia is usually stored under slightly lower temperature than ambient conditions (240 K), or 10 bar atmospheric [2,3]. Because of this low temperature, ammonia can cause frostbite [107]. All of this combined, it is expected that the MLC will require certain measures to mitigate the risk that ammonia poses to the crew on board, as having ammonia on board is dangerous for the crew and passengers.

The above-mentioned hazards related to life on board are summarized in Figure 4. Because of the large differences between AB and its reaction products, this is split up in three types. It should again be noted that this is a qualitative assessment and the numbers are arbitrary, it should only give a comparison compared to each other and to MDO.

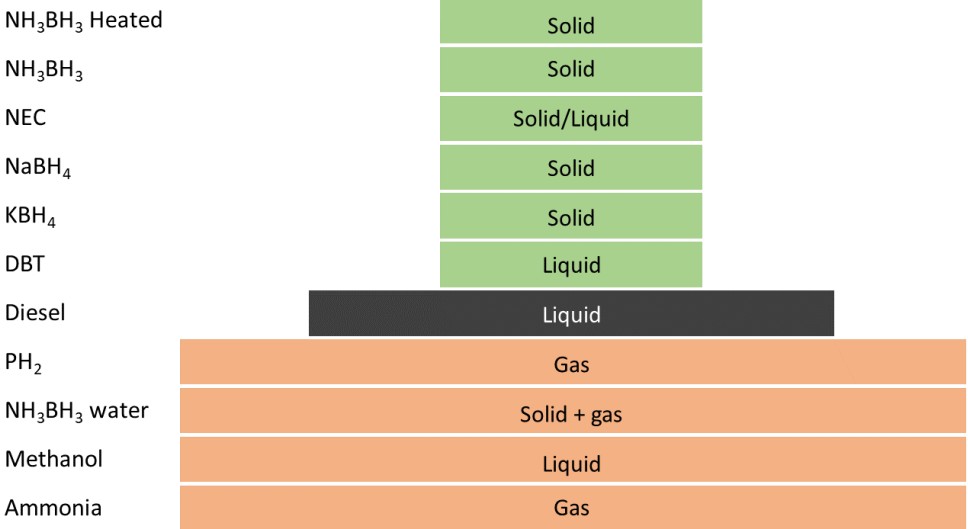

**Figure 4.** Qualitative comparison of hazards accompanied by future fuels and MDO relevant for the safety of crew and passengers. Here, only hazards are taken into account, no probabilities or resulting risks. Because of the different behaviors of substances due to their state, the state is taken into account.

*5.2. Possible Influence of Alternative Fuels on Ship Structural Integrity*

The functional safety and safety aspects regarding structural integrity of the ship are influenced by two GHS aspects of the hydrogen carriers studied: flammability and hydrogen explosions. Within the GHS [34], flammability is (almost) always considered a hazard of danger level instead of a warning level. Two fuels are flammable without additional requirements: AB and methanol. Methanol burns with an invisible flame, resulting in the requirement for additional detection equipment. Therefore, only AB forms a direct threat to ship integrity based on it's flammability properties. Since conventional fuels share the same flammability properties, fire on board is a threat that is well-considered and included in the IMO Conventions. Additionally, $NaBH_4$ and $KBH_4$ also have flammable properties when in contact with water due to the release of hydrogen. The quantity of hydrogen released without a catalyst is limited, but still to be considered onboard ships. Finally, influence of hydrogen carriers in general on materials still have to be researched.

It is trivial that hydrogen is present at one stage in the process since it must be extracted from hydrogen carriers. Therefore, hazards related to hydrogen are applicable to all hydrogen carriers considered. One of these hazards has been mentioned in detail in Section 3.8.1. Other hazards, such as permeation and the influence of hydrogen on other materials, are recommended for further research.

*5.3. Possible Influence of Alternative Fuels on the Environment*

Environmental hazards are always considered as a warning level threat instead of a danger level threat. It is also possible for environmental hazards to only have a pictogram, without a signal word [34]. The four fuels that, according to literature, pose a threat to the environment are DBT, AB, $NH_3$ and MDO, with AB only after contact with water. DBT forms a hazard in the long term only, as it is extremely toxic to aquatic life in the long term. It is not persistent, with a high biodegradability. NEC on the other hand, is not biodegradable, but is also thought to be not acutely toxic. The exact consequence of this persistence is still to be studied, but there is a common understanding that it causes an environmental hazard. Especially when compared to MDO, this poses different hazards, as MDO is biodegradable [41]. AB is only toxic to aquatic life if it hydrolyzes, as then ammonia is released. Hydrolysis cannot be excluded here as it is likely to happen over time, despite AB being relatively stable in water. Ammonia is not only toxic to humans, but also to aquatic life. Moreover, contrary to conventional fuels, it cannot be flared to remove it from a marine environment. Similarly, AB cannot be flared because it is highly soluble in water and thus will not ignite despite its high flammability. Thus, it is hard to compare the effect on the aquatic environment due to the different types of effects of the alternative fuels. The MARPOL [96] has appendices on oil spillage, which forms the main threat for environment by MDO. The spillage of ammonia, DBT and NEC is not mentioned in MARPOL as a risk to environment [108]. This shows that MARPOL has the potential to improve in the area concerning alternative fuels.

## 6. Conclusions

The discussed hydrogen carriers cannot be called safe by definition based on their GHS classifications as defined in Section 3. Moreover, the safety of hydrogen carriers is not fully integrated in the IMO safety framework. The integration of the hydrogen carrier chemical safety with the maritime safety approach is limited due to the following characteristics. First, the maritime safety definition often lacks technical justification and the chemical safety categorization is non-explicit. Therefore, matching these definitions induces an even greater level of uncertainty causing a fully subjective ranking of hydrogen carriers on their maritime safety level. The GHS classifications are quantitative, with only three classes: nothing, warning or danger. Even though these classifications are based on underlying criteria, these criteria are still divided in four hazard levels. The IMO classification of safety is found to be even more ambiguous, with either qualitative and abstract goal-based standards or prescriptive standards solely based on common practice with fossil fuels. This combination makes it challenging to assess hydrogen carriers on their safety level within a maritime context. It is noteworthy that MDO is also categorized as danger for multiple GHS labels. An attempt to connect the GHS labels to the three previously mentioned maritime safety aspects is shown in Figure 5. The three aspects are, respectively, safety of life on board, safety of ship structural integrity and safety of ecology. The "Warning" and "Danger" values are in line with Table 2.

Overall, it cannot be said that there is one hydrogen carrier that is the safest. The only real conclusion that can be made is whether substances have or have no influence on the certain safety aspects, but even this conclusion can change with design and mitigation.

*Discussion and Further Research*

In order to draw strong conclusions on which hydrogen carriers are safer than others, future research should focus on ways of comparing qualitative safety values as well on mitigative or preventative measures. There are several frameworks of comparing qualitative and quantitative values to guide designers in making decisions when values may conflict or are hard to measure, as shown by [109]. Some of these conflicts may also disappear when safety is incorporated in the beginning. An example of designing for safety can be found in the framework of safe-by-design [110]. Comparing these strategies can result in a prioritization of different values within a maritime context. This way, a

considered, defensible deliberation can be made for different design aspects as safety, mass, volume and cost.

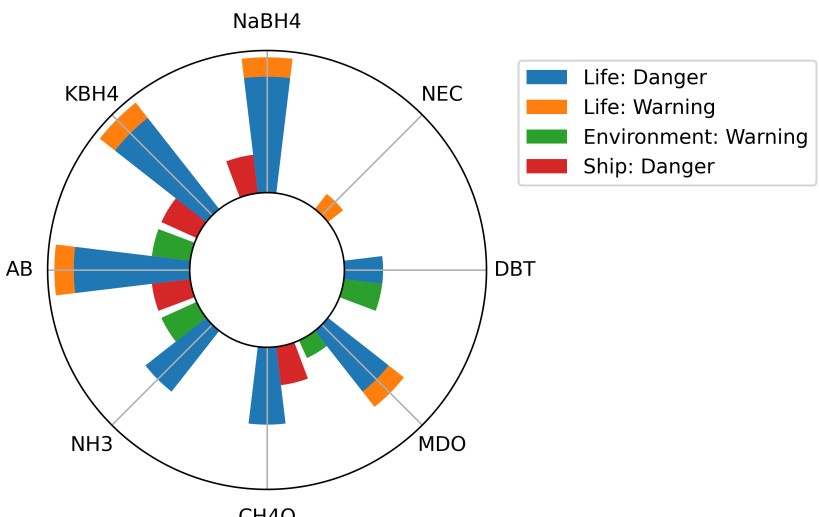

**Figure 5.** An overview of the alternative fuels and their hazards based on the three major focal points: life of on board (life), structural integrity of the ship (ship) and the aquatic environmental impact (environment), including the dangers and warnings stacked together. The bar height is based on the number of hazards per category, with D having more weight than W.

**Author Contributions:** Conceptualization, E.v.R., E.S. and K.V.; Data curation, E.v.R., E.S. and J.Z.; Funding acquisition, K.V.; Investigation, E.v.R. and J.Z.; Methodology, E.v.R., E.S. and J.Z.; Resources, E.S., J.Z. and K.V.; Supervision, E.v.R. and K.V.; Visualization, E.v.R. and E.S.; Writing—original draft, E.v.R., E.S. and J.Z.; Writing—review and editing, E.v.R., E.S., J.Z. and K.V. All authors have read and agreed to the published version of the manuscript.

**Funding:** This work was supported by the project SH2IPDRIVE, which has received funding from the Ministry of Economic Affairs and Climate Policy, RDM regulation, carried out by the Netherlands Enterprise Agency (RvO) and the project READINESS with project number TWM.BL.019.002 of the research program "Topsector Water and Maritime: the Blue route" which is partly financed by the Dutch Research Council (NWO).

**Institutional Review Board Statement:** Not applicable.

**Informed Consent Statement:** Not applicable.

**Data Availability Statement:** No new data were created or analyzed in this study. Data sharing is not applicable to this article.

**Conflicts of Interest:** The authors declare no conflicts of interest. The funders had no role in the design of the study; in the collection, analyses, or interpretation of data; in the writing of the manuscript; or in the decision to publish the results.

## Abbreviations

The following abbreviations are used in this manuscript:

| | |
|---|---|
| AB | Ammoniaborane, $NH_3BH_3$ |
| $B(OH)_4^-$ | Tetrahydroxyboranuide |
| $BO_2$ | Metaborate |
| $CO_2$ | Carbon dioxide |
| D | Danger |
| $D_{hmsc,c}$ | Diameter of a hydrogen fireball, conservative estimation |
| DBT | 2,3-dibenzyltoluene |
| GBS | Goal-based standards |

| | |
|---|---|
| GHG | Greenhouse gas emissions |
| GHS | Globally Harmonized System of classification and labelling of chemicals |
| (H$_3$B$_3$N$_3$H$_3$) | Borazine |
| HFO | Heavy fuel oil |
| IGF | International code of safety for ships using gases or other low-flashpoint fuels |
| ILO | International Labor Organization |
| IMO | International Maritime Organization |
| KBH$_4$ | Potassium borohydride |
| KBO$_2$ | Potassium metaborate |
| LNG | Liquid natural gas |
| LOHC | Liquid organic hydrogen carrier |
| UNCLOS | United Nations Convention on the Law of the Sea |
| MARPOL | International convention for the prevention of pollution from ships |
| MDO | Marine diesel oil |
| MEPC | Marine Environment Protection Committee |
| MLC | Maritime Labour Convention |
| MSC | Maritime Safety Committee |
| NaB(OH)$_4$ | Sodium metaborate dihydrate |
| NaBH$_4$ | Sodium borohydride |
| NaBO$_2$ | Sodium metaborate |
| NEC | N-ethylcarbazole |
| NH$_2$BH$_2$ | Aminoborane |
| NH$_3$ | Ammonia |
| NH$_4^+$ | Ammonium |
| NHBH | Iminoborane |
| NO$_x$ | Nitrogen oxides |
| SOLAS | International Convention for the Safety of Life at Sea |
| SO$_x$ | Sulphur oxides |
| TPI | Toxicity probability interval |
| TRL | Technology readiness level |
| W | Warning |

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
