# Peer review of "Hazard Identification of Hydrogen-Based Alternative Fuels Onboard Ships"

_sustainability, doi:10.3390/su152416818_

Round 1

Reviewer 1 Report

Comments and Suggestions for Authors

The author presents a Hazard Identification of Hydrogen-Based Alternative Fuels Onboard Ships. The paper is interesting, complete and easy to follow. I have a few comments as follows:

1. The authors can provide more information about this review paper compared to the published one, showing the novelty of the work.

2. Some ref are missed as in sec 3.3

3. Does the hazard identification differ if different types of fuel are on-board? Please elaborate.

4. The conclusion can be improved.

5. The authors can provide more generic procedures for hazard identification and assessment and how the different refs can achieve that.

6. What is the recommendation from this paper to ensure the safety of the ships.

Author Response

Dear reviewer,

Thank you for your elaborate feedback. Please see the attached document for our responses and reactions.

Kind regards,

Erin van Rheenen

Reviewer 2 Report

Comments and Suggestions for Authors

The paper reviews the alternative fuels used in the maritime industry along with the safety issues which accompanies their utilization. The safety protocols/regulations were also reviewed to highlight the need to improve on the current standards to encompass holistic safety issues which should include structural integrity and ecology.

Whilst the manuscript appears to be well thought out, some comments will improve the manuscript:

1. Define abbreviations at their first use within the manuscript. This is despite defining them in the list of abbreviations. It will aid readability of the manuscript.

2. Correct typographical errors, grammatical errors and punctuation errors throughout the manuscript. Also check the use of prepositions.

3. Recheck the sentences in Line 126-127; 170-173.

4. The conclusion in Line 190-191 does not seem to follow from the review within the section.

5. Check Line 196. What do the question marks mean?

6. Line 204: Conclusion should be based on your observations from the review, but you may then add that further hazard assessment is required.

7. The sentence in Line 219 looks incomplete.

8. Line 222, 514: What are restproducts? Do you mean byproducts?

9. Line 494, I think the section title should be modified by changing “accompanied by” to “accompanying”

10. Change “section” to “Section” throughout the manuscript.

11. Line 636: What are the 3 classes? Does it include “nothing” or “low”

12. The title of Section 6.1 should be removed and Section 6 may be titled “Conclusion and Further Research”

Comments on the Quality of English Language

Correct typographical errors, grammatical errors and punctuation errors throughout the manuscript. Also check the use of prepositions.

Author Response

(The authors gave the same response as above.)
